# suddengains: An R package to identify sudden gains in longitudinal data

**Milan Wiedemann**[1,2]*, **Graham R. Thew**[1,2,3], **Richard Stott**[4], **Anke Ehlers**[1,2,4]

**1** Department of Experimental Psychology, University of Oxford, Oxford, United Kingdom, **2** Oxford Health NHS Foundation Trust, Oxford, United Kingdom, **3** Oxford University Hospitals NHS Foundation Trust, Oxford, United Kingdom, **4** King's College London, London, United Kingdom

* milan.wiedemann@gmail.com

## Abstract

Sudden gains are large and stable improvements in an outcome variable between consecutive measurements, for example during a psychological intervention with multiple assessments. Researching these occurrences could help understand individual change processes in longitudinal data. Three criteria are generally used to identify sudden gains in psychological interventions. However, applying these criteria can be time consuming and prone to errors if not fully automated. Adaptations to these criteria and methodological decisions such as how multiple gains are handled vary across studies and are reported with different levels of detail. These problems limit the comparability of individual studies and make it hard to understand or replicate the exact methods used. The R package *suddengains* provides a set of tools to facilitate sudden gains research. This article illustrates how to use the package to identify sudden gains or sudden losses and how to extract descriptive statistics as well as exportable data files for further analysis. It also outlines how these analyses can be customised to apply adaptations of the standard criteria. The *suddengains* package therefore offers significant scope to improve the efficiency, reporting, and reproducibility of sudden gains research.

## Introduction

A sudden gain is a large improvement in an outcome variable experienced by an individual participant between two consecutive measurement points that is stable within a longitudinal data series. Sudden gains were first defined and investigated by Tang and DeRubeis [1], who examined session to session changes in depression symptoms among participants undertaking cognitive behavioural therapy. The majority of sudden gains studies to date have been in relation to psychological therapies [2], but the analytic approach could also be considered when investigating within-participant changes in other fields. A meta-analysis of 16 studies of psychological therapies (total $N = 1104$) found that experiencing a sudden gain was associated with better overall clinical outcomes at the end of treatment and at follow-up compared to those who did not experience gains [2]. Given this potential significance of sudden gains, examining such events specifically may be informative in understanding when and why such

**Data Availability Statement:** The package can be downloaded from CRAN https://CRAN.R-project.org/package=suddengains. All code, materials, and data can be found at https://github.com/milanwiedemann/suddengains. Instructions for

installing the package, further technical details, and examples can be found at https://milanwiedemann. github.io/suddengains. The R code examples in this paper refer to package version 0.4.0.

**Funding:** This project was supported by a Mental Health Research UK studentship (MW), the Wellcome Trust [102176 (GRT); 069777 and 200976 (AE, RS)], the Oxford Health NIHR Biomedical Research Centre (MW, GRT, AE), and the NIHR Oxford Biomedical Research Centre (GRT). The views expressed are those of the authors and not necessarily those of the NHS, the NIHR or the Department of Health. The funders had no role in study design, data collection and analysis, decision to publish, or preparation of the manuscript.

**Competing interests:** The authors have declared that no competing interests exist.

large improvements occur, which could help to increase the efficacy and efficiency of the intervention.

Rates of sudden gains within published clinical studies vary considerably (e.g. 17.8% to 52.2% of participants [2]), which may partly be due to differences in the methods used to identify them. However, such differences are hard to examine given that sufficient methodological details to permit a comparison are not always reported. In addition, some studies have raised concerns about the validity of sudden gains identified through current methods, demonstrating that they can be found in placebo interventions and simulated datasets [3, 4]. This suggests that not all gains reflect meaningful change or show a causal association with the intervention being studied. This highlights the need to examine the presence and strength of these associations and to consider if the current methods of identification can be refined The *suddengains* R package is the first software program to offer explicit and reproducible methods to automatically identify sudden gains, which may be valuable in improving methodological reporting and consistency across studies. It may also facilitate closer examination of the methods used to identify sudden gains, to help improve their validity and ensure that they more accurately reflect meaningful events. This article aims to provide an accessible overview of how sudden gains are calculated, describe the principal functions of the package, and give instructions on how to use these with longitudinal data. It is hoped that using this package will facilitate improvements in the efficiency, reporting, and reproducibility of sudden gains research.

## Identification of sudden gains

Tang and DeRubeis [1, 5] suggested the following three criteria to identify sudden gains:

1. The gain must be large in absolute terms. While this was originally operationalised as a decrease of at least 7 points on the Beck Depression Inventory (BDI [6]), subsequent studies have generally used the Reliable Change Index (RCI [7]) to define an appropriate cutoff for other scales [8]. Further details are discussed below.

2. The gain must be large in relative terms. This is defined as a drop of at least 25% of the previous score.

3. The gain must be large relative to symptom fluctuation. Originally an independent *t* test was proposed to compare the size of the sudden gain with symptom fluctuation before and after the gain. This method was controversial given the assumption of independence of the measurements before and after the gain is not met [3, 9]. Consequently the wording of this criterion was updated by Tang and colleagues [5, 9], though the calculations remained the same: The difference between the mean scores of the three measurements before the gain ($M_{\text{pre}}$), and the three measurements after the gain ($M_{\text{post}}$), must be greater than the pooled standard deviation of these two groups multiplied by a critical value of 2.776 (i.e. the two-tailed *t* statistic for $\alpha = 0.05$ and $df = 4$). The formula for criterion 3 is therefore:

$$M_{pre} - M_{post} > \text{critical value} * \sqrt{\frac{(n_{pre} - 1) * SD^2_{pre} + (n_{post} - 1) * SD^2_{post}}{n_{pre} + n_{post} - 2}} \qquad (1)$$

The criteria used to identify sudden gains vary between studies. For example, some studies have used different methods to define a cutoff value for criterion one [10, 11], criterion two was not included in some studies because of concerns about the impact of different response scales and data suggesting it has little effect on the number of gains found [12], and studies have used different methods to select a critical value for use in criterion 3 [11, 13] see Eq 1.

## Defining a cutoff for the first criterion

Tang and DeRubeis [1] originally defined a 7 point cutoff on the BDI for the first criterion based on frequency distribution plots of session to session change scores on the BDI in clinical trials. The authors reported that 7 BDI points approximately reflected one standard deviation in clinical samples [9]. Stiles et al. [8] noted that 7 BDI points was close to the reliable change value reported in Barkham et al. [14] and therefore used the RCI formula to define a cutoff for a new measure. Subsequent studies have generally adopted this approach. Jacobson and Truax [7] proposed the following formula to test whether the observed pre to post change on a measure reflects more than just fluctuation due to measurement error:

$$\frac{\text{pre} - \text{post}}{S_{\text{diff}}} = \text{RCI} \tag{2}$$

Following Jacobson and Truax [7], reliable change on a measure is present when:

$$\frac{\text{pre} - \text{post}}{S_{\text{diff}}} > 1.96; \text{ therefore} \tag{3}$$

$$\text{reliable change} > 1.96 \times S_{\text{diff}}; \tag{4}$$

where $S_{\text{diff}}$ is the standard error of the difference between pre and post scores. Using the standard error of measurement ($S_E$), $S_{\text{diff}}$ can be expressed as:

$$S_{\text{diff}} = \sqrt{2 \times (S_E)^2}; \tag{5}$$

where $S_E$ is calculated using the standard deviation of the control group or normal population $s_1$ and the test-retest reliability of the measure ($r_{xx}$):

$$S_E = s_1 \sqrt{1 - r_{xx}}; \tag{6}$$

Some studies have adapted this formula following suggestions from Martinovich, Saunders and Howard [15] by replacing the test-retest reliability with the internal consistency ($\alpha$) and replacing the standard deviation of the normal population ($s_1$) with the standard deviation of the clinical sample at baseline ($SD_{\text{pre}}$) so that all statistics can be extracted from the sample data [16]. Note that the use of the test-retest reliability or internal consistency when calculating $S_E$ makes the assumption that the scale being examined is unidimensional, and that these reliability estimates remain constant over time, and between individuals. Exploring the factor structure and measurement invariance of the scale may be appropriate to examine if these assumption hold.

$$S_E = SD_{pre}\sqrt{1 - \alpha} \tag{7}$$

In the sudden gains literature different approaches have been used to define a cutoff for the first criterion using the RCI formula. Some studies [10, 17] have used the standard error of the difference ($S_{\text{diff}}$) while others [11, 13] have used the reliable change value ($1.96 \times S_{\text{diff}}$). When defining a cutoff it is important to consider the statistical assumptions involved, and toensure that this value reflects a meaningful change (large in absolute terms) that is realistic in a session by session context for the intervention.

## Missing data

Missing data, for example where a participant does not provide data on one or more occasions, need to be considered carefully when identifying sudden gains for several reasons. Firstly, depending on the number and pattern of missing data points for an individual, it may not be possible to identify sudden gains, see Table 1. Specifically, in order to estimate the standard deviation values in criterion 3, at least two of the three measurements immediately prior to the gain must be present, as well as at least two of the three measurements immediately following the gain. Some researchers have suggested that methods used to replace missing values, such as *last observation carried forward* or *multiple imputation*, may not be appropriate when identifying sudden gains given the potential for additional gains to be detected based on data that were not provided by participants [18, 19].

Secondly, where values are missing in the period around the potential sudden gain, two approaches have been described to evaluate the stability of the change. Following the updated version of the third criterion by Tang and colleagues [5, 9] some studies have used a critical value of 2.776 across all session to session intervals to check the stability [13]. An alternative approach adjusts the critical values used in criterion 3 (see Eq 1) based on the data that were available in the period around the potential sudden gain [11]: Where no data are missing $t_{(4;97.5\%)} > 2.776$; where one datapoint is missing either before or after the gain $t_{(3;97.5\%)} > 3.182$; and where one datapoint is missing both before and after the gain $t_{(2;97.5\%)} > 4.303$. This method has been adopted in some subsequent studies [20, 21].

It is important to understand the reasons for missing data and consider whether methods to handle missing data need to be employed both at the identification stage and in subsequent analyses [22, 23]. Further research to examine the impact of missing data and different methods to handle missing data when identifying sudden gains would be beneficial.

## Terminology

The naming of specific sessions (or measurement points) around the gain follows the convention that the session immediately prior to the gain is session *N* (also known as the *pregain* session), and the session immediately after is session *N+1* (or *postgain* session). Other sessions are referred to in relation to session *N* (e.g. *N-2*, *N+3*).

## Reversals

According to Tang and DeRubeis [1] a sudden gain is counted as reversed if 50% of the improvement made during the gain was lost at any subsequent point. For example, where the sudden gain represents a drop in score from 40 to 30 points, the gain is classed as having

**Table 1. Data patterns required to identify sudden gains.**

|            | $x_{n-2}$ | $x_{n-1}$ | $x_n$ | $x_{n+1}$ | $x_{n+2}$ | $x_{n+3}$ |
|------------|-----------|-----------|-------|-----------|-----------|-----------|
| Pattern 1  | ○         | •         | •     | •         | •         | ○         |
| Pattern 2  | ○         | •         | •     | •         | ○         | •         |
| Pattern 3  | •         | ○         | •     | •         | •         | ○         |
| Pattern 4  | •         | ○         | •     | •         | ○         | •         |

*Note.* $x_{n-2}$ to $x_{n+3}$ represent any six consecutive measurement points within the data set. The minimum number of data points that must be present (•) in order to investigate the interval from $x_n$ to $x_{n+1}$ as a potential sudden gain is four, arranged in one of the patterns shown. Note that the pregain ($x_n$) and postgain ($x_{x+1}$) data points must always be present. ○ represents missing data.

reversed if a score of 35 or more is observed at any later session. As discussed in Wucherpfennig et al. [20] a reversal might not necessarily be a stable phenomenon. These authors modified this criterion by suggesting that a stable reversal is present when a reversal is also classified as a sudden loss (see below).

### Sudden losses

Although less frequently studied than sudden gains, sudden losses represent the inverse phenomenon, where a participant shows a large and stable increase of scores on the outcome variable. While some authors invert the three sudden gains criteria [11, 24], others further adjust the percentage threshold of the second criterion, e.g. 33% [16].

### Why is a package needed?

As indicated by the criteria above, identifying sudden gains requires the application of each of the three criteria to each session to session interval, and that this is performed for each individual in a given dataset. A large number of calculations and extensive manipulation of data is therefore involved, particularly in larger datasets. Doing these data manipulations manually (e.g. in spreadsheets) can be extremely time consuming and lead to errors. It also means that certain methodological decisions, such as determining the critical value for the third criterion, or handling of participants with multiple gains, may not be addressed sufficiently or in a consistent way across studies. It is hoped that the use of the *suddengains* package will provide faster and more accurate calculations, as well as offering a transparent and consistent method to address these methodological considerations.

### Functions of the *suddengains* package

The *suddengains* package provides a set of functions to calculate the presence of sudden gains (and sudden losses) within a longitudinal dataset, and to provide basic plots and descriptive statistics of the gains. It can also extract scores on secondary outcome or process measures around the period of each gain. Output files (in SPSS, Excel, or CSV formats) arranged by individual gain, or by person can be generated for further analyses in other programs. This package is supplemented by an interactive web application [25] *shinygains* that illustrates the main functions of this package at https://milanwiedemann.shinyapps.io/shinygains/. As it allows users to explore and understand the impact of different methodological choices, it may be useful in planning sudden gains studies. Table 2 lists and describes the main functions.

### Worked example

This demonstration uses a dataset (`sgdata`) that was created to illustrate the functions of this package. The data show self-report weekly questionnaire scores for 43 participants who have received psychological therapy for depression. The intervention lasted for 12 sessions, and each participant completed a set of outcome measures at the beginning of each session, including the BDI and a fictional secondary measure assessing rumination (RQ).

### Preparation of data

The data to be analysed for sudden gains are arranged in wide format i.e. one row per participant, and one column for each questionnaire score at each measurement point. A unique identifier variable also needs to be included. Some researchers have specified a minimum number of measurement points that must be present for participants to be included, to ensure that they received a sufficient amount of the intervention being studied [1]. Alternatively it may be of

**Table 2. Main functions of the *suddengains* R package.**

| Function | Description |
|---|---|
| **Identify sudden gains** | |
| `define_crit1_cutoff()` | Uses RCI formula to help determine a cutoff value for criterion 1 |
| `check_interval()` | Checks if a given interval is a sudden gain/loss |
| `identify_sg(),identify_sl()` | Identifies sudden gains/losses |
| **Create datasets** | |
| `create_bysg(),create_byperson()` | Creates a dataset with one row for each sudden gain/loss |
| `extract_values()` | Extracts values on a secondary measure around the sudden gain/loss |
| **Describe sudden gains** | |
| `describe_sg()` | Generates summary descriptive statistics |
| `plot_sg(),plot_sg_trajectories()` | Creates plots of the average sudden gain, or individual case trajectories |
| **Additional functions** | |
| `select_cases()` | Selects cases to be included in the sudden gains analysis |
| `write_bysg(),write_byperson()` | Exports CSV, SPSS, Excel, or STATA files of the sudden gains datasets |

*Note*. More details of each function can be found in the package documentation or using the help() function in R.

interest to analyse all cases whose data are distributed such that at least one interval can be examined for a potential sudden gain [21]; For all three criteria to be applied there must be data present for at least two of the three data points prior to, and two of the three following, the interval to be examined, see Table 1. The optional `select_cases()` function can be used to identify samples of cases for analysis who fulfil such conditions, though researchers should consider whether these methods are appropriate for the aims of the study.

## Identification of sudden gains

The `identify_sg()` function applies the sudden gains criteria as specified by the user to each session to session interval in the dataset. As shown below, the user specifies: `data`, the dataset to use in wide format; `sg_crit1_cutoff`, the cutoff value to use for criterion 1 (which can be entered manually or calculated using the `define_crit1_cutoff()` function); `sg_crit2_pct`, the percentage change value to use for criterion 2 (0.25 by default); `sg_crit3`, whether or not to apply the third criterion (`TRUE` by default); `sg_crit3_alpha`, the alpha value to use when calculating the criterion 3 critical value (0.05 by default); `id_var_name`, the name of the unique identifier variable within the dataset; and `sg_var_list`, a list of the variables representing the span of sessions to be analysed, which is sessions 1 to 12 in this example. By default all functions that identify sudden gains apply the adjustment of the critical value in Eq 1 as described by Lutz and colleagues [11]. To turn off this adjustment and instead apply a manually defined critical value across all session to session intervals, the argument `sg_crit3_adjust = FALSE` can be included and `sg_crit3_critical_value` specified. Additional options to customise this analysis are discussed in the package documentation. An alternative function, `identify_sl()`, is identical to `identify_sg()` but applies the criteria in the inverse direction to calculate sudden losses. The function `check_interval()` can be used to examine whether a specific session to session interval is a sudden gain/loss.

**Table 3. A sample of the output data frame created by the `identify_sg()` function.**

| id | sg_crit1_2to3 | sg_crit2_2to3 | sg_crit3_2to3 | sg_2to3 |
|----|---------------|---------------|---------------|---------|
| 1 | FALSE | FALSE | FALSE | 0 |
| 2 | *NA* | *NA* | *NA* | *NA* |
| 10 | TRUE | TRUE | TRUE | 1 |
| 12 | TRUE | FALSE | FALSE | 0 |
| 18 | FALSE | FALSE | *NA* | *NA* |
| 23 | FALSE | FALSE | TRUE | 0 |

*Note*. For the variables testing the three sudden gains criteria, referred to by 'crit1', 'crit2', and 'crit3' in the variable names TRUE indicates that the criterion is met, while FALSE indicates the criterion is not met. *NA* indicates that a particular criterion could not be tested for a sudden gain due to missing data.

```
# First, install and load the suddengains R package
install.packages("suddengains")
library(suddengains)
# Identify sudden gains in the dataset "sgdata":
identify_sg(data = sgdata,
        sg_crit1_cutoff = 7,
        sg_crit2_pct = 0.25,
        sg_crit3 = TRUE,
        id_var_name = "id",
        sg_var_list = c("bdi_s1", "bdi_s2", "bdi_s3", "bdi_s4",
                "bdi_s5", "bdi_s6", "bdi_s7", "bdi_s8",
                "bdi_s9", "bdi_s10", "bdi_s11", "bdi_s12"),
        crit123_details = TRUE)
```

The output data frame shows each session to session interval, for example `sg_2to3` representing the interval between sessions two and three. Variables indicate whether each of the three criteria were met and therefore whether a sudden gain was observed for each interval. Sudden gains are indicated by a value of 1, see Table 3. Examining this interval in our example data, we see that only `id = 10` meets all three criteria, for `id = 2` none of the three criteria can be tested, for `id = 18` only the third criterion can not be tested, for all other participants at least one criterion is not met.

To permit further analysis of our data, we wish to obtain an output dataset containing both the original data and the newly identified sudden gains. As participants may experience more than one gain, as in the present example, and to allow for different subsequent analyses, the package provides two options for output datasets: The `create_bysg()` function creates a dataset structured with one row per sudden gain, and the `create_byperson()` function creates a dataset structured with one row per person, indicating whether or not they experienced a sudden gain. The `tx_start_var_name` and `tx_end_var_name` arguments are used to specify the start and end of treatment (tx) variables, and `sg_measure_name` specifies the name of the measure used to calculate sudden gains.

```
# Create output dataset with one row per sudden gain
# and save as an object called "bysg" to use later
bysg <- create_bysg(data = sgdata,
            sg_crit1_cutoff = 7,
            sg_crit2_pct = 0.25,
            sg_crit3 = TRUE,
            id_var_name = "id",
            tx_start_var_name = "bdi_s1",
            tx_end_var_name = "bdi_s12",
            sg_var_list = c("bdi_s1", "bdi_s2", "bdi_s3",
                    "bdi_s4", "bdi_s5", "bdi_s6",
```

```
              "bdi_s7", "bdi_s8", "bdi_s9",
              "bdi_s10", "bdi_s11", "bdi_s12"),
         sg_measure_name = "bdi")
```

The new variables created by the `create_bysg()` and `create_byperson()` functions are described in Table 4. To continue working in another program (e.g. SPSS, STATA, Excel) the functions `write_bysg()` and `write_byperson()` can be used to export the datasets created in R [26] as *.sav*, *.dta*, *.xlsx*, or *.csv* files.

## Analysis of sudden gains

In this example, we have calculated sudden gains based on depression scores using the BDI. In analysing these gains, we are interested in how rumination scores on the fictional RQ measure change around the period of the sudden gains in depression. The `extract_values()` function extracts the RQ values from the three sessions before (*N-2*, *N-1*, *N*) and the three sessions after (*N+1*, *N+2*, *N+3*) each depression sudden gain. In the dataset that gets returned by this function we refer to these sessions as sg_bdi_**2n**, sg_bdi_**1n**, sg_bdi_**n**, sg_bdi_**n1**, sg_bdi_**n2**, and sg_bdi_**n3**, respectively. This function can be applied to either the `bysg` or `byperson` dataset. By default the extracted values will be added as new variables to the dataset used. Here we demonstrate applying this function to the `bysg` dataset, as shown in the code below. First, the RQ variables are added to the `bysg` dataset. Second, the `extract_values()` function is applied. Note that the list of RQ variables included in the `extract_var_list` argument must match those used for the `sg_var_list` argument used previously in the `create_bysg()` function. This means that the number of variables in these lists has to be identical and measured at the same timepoints. The output data frame can be saved as a new object, or the existing `bysg` object can be overwritten, as in this example. The RQ scores now in the `bysg` dataset can be examined, for example to look at the temporal relationship between changes in rumination and changes in depression symptoms.

**Table 4. Description of variables created by the `create_bysg()` and `create_byperson()` functions.**

| Variable Name | Variable Label |
|---|---|
| id_sg | Unique ID variable for every identified sudden gain / loss |
| sg_crit123 | Indicates whether all applied sudden gain criteria were met (No = 0; Yes = 1) |
| sg_session_n | Pregain session number |
| sg_freq_byperson | Frequency of sudden gains / losses per person |
| sg_bdi_2n | Pre-pre-pre gain session score (N-2) |
| sg_bdi_1n | Pre-pre gain session score (N-1) |
| sg_bdi_n | Pre-gain session score (N) |
| sg_bdi_n1 | Post-gain session score (N+1) |
| sg_bdi_n2 | Post-post gain session score (N+2) |
| sg_bdi_n3 | Post-post-post gain session score (N+3) |
| sg_magnitude | Raw magnitude of sudden gain |
| sg_bdi_tx_change | Total change during treatment |
| sg_change_proportion | Proportion of total change represented by the sudden gain |
| sg_reversal_value | Reversal value |
| sg_reversal | Indicates whether the reversal value was met at any point in treatment following the sudden gain (No = 0; Yes = 1) |

*Note*. The variable names listed including _bdi_ will reflect the name of the measure specified in the `sg_measure_name` argument.

```
# 1. Select the ID and variables from a second measure
sgdata_rq <- dplyr::select(sgdata,
                "id",
                "rq_s1", "rq_s2", "rq_s3",
                "rq_s4", "rq_s5", "rq_s6",
                "rq_s7", "rq_s8", "rq_s9",
                "rq_s10", "rq_s11", "rq_s12")
# 2. Add variables in 'sgdata rq' to the 'bysg' dataset created
earlier
bysg <- dplyr::left_join(bysg, sgdata_rq, by = "id")
# 3. Extract values on the second measure around the sudden gain
bysg <- extract_values(data = bysg,
                id_var_name = "id_sg",
                extract_var_list = c("rq_s1", "rq_s2", "rq_s3",
                        "rq_s4", "rq_s5", "rq_s6",
                        "rq_s7", "rq_s8", "rq_s9",
                        "rq_s10", "rq_s11", "rq_s12"),
                extract_measure_name = "rq",
                add_to_data = TRUE)
```

The `describe_sg()` function provides descriptive statistics about the sudden gains based on the variables from the `bysg` and `byperson` datasets. For the present example, this function indicates that 16 of the 43 participants experienced a sudden gain, and 9 experienced more than one gain, leading to a total of 26 sudden gains within the data. Information on the mean gain magnitude and reversals is also provided.

The `plot_sg()` function plots the average sudden gain, and can be used to show the primary or secondary outcome measure data (Fig 1A and 1B). The `sg_pre_post_var_list` argument specifies the pregain and postgain variables to be plotted, namely sessions *N-2* to *N+3*. This function is built using the R Package *ggplot2* [27] and additional *ggplot2* functions can be added to the plot. It is also possible to plot the average gain magnitude of different groups (e.g. two treatment arms in a trial) in one figure by using the optional `group` argument (see Fig 1C).

```
# Create average sudden gain plot for BDI data (see Fig 1A):
plot_sg(data = bysg,
     id_var_name = "id",
     tx_start_var_name = "bdi_s1",
     tx_end_var_name = "bdi_s12",
     sg_pre_post_var_list = c("sg_bdi_2n", "sg_bdi_1n", "sg_bdi_n",
                     "sg_bdi_n1", "sg_bdi_n2", "sg_bdi_n3"),
     ylab = "BDI")
```

An additional function, `plot_sg_trajectories()`, is available to plot the trajectories of a selection of individual cases within the dataset (see Fig 2A). This function can be paired with a filter command, for example `filter()` from the R Package *dplyr* [28], to visualise trajectories of specific groups of participants. For example, all participants with more than one sudden gain, or all participants with a sudden gain between sessions 3 and 4 (see Fig 2B).

## Discussion

The analysis of sudden gains and losses provides a detailed examination of within-participant changes during the course of an intervention, and may help to understand individual processes of change. The *suddengains* package aims to facilitate the computation of gains, which can be laborious and error-prone. It also aims to address common methodological issues, for example by allowing adjustments to the critical value for the third criterion in the presence of missing data, and by highlighting participants with multiple gains.

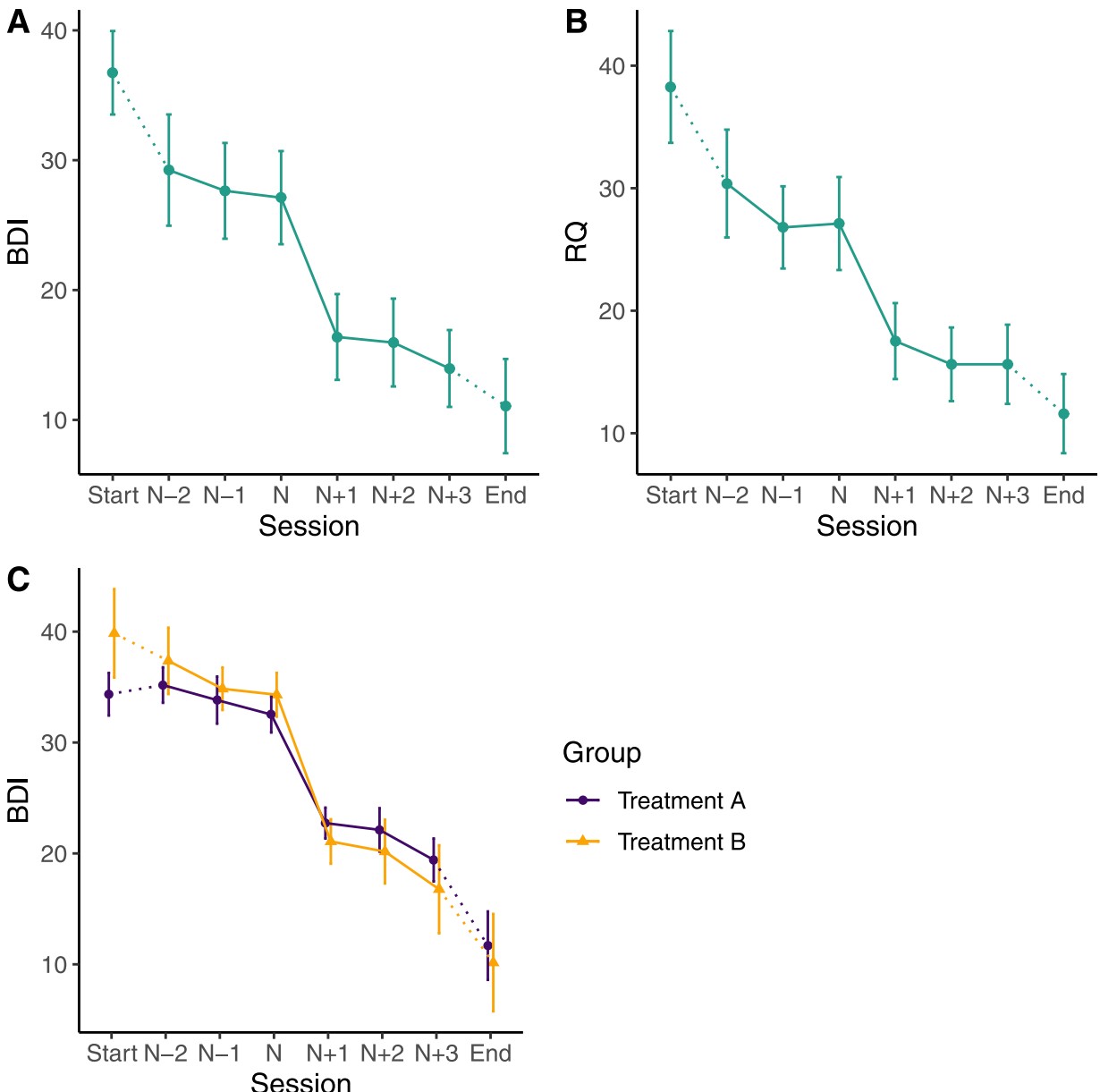

**Fig 1. Plots of average changes around sudden gains.** (A) Average gain magnitude on the BDI across all sudden gains (B) Average changes in rumination (RQ) around sudden gains on the BDI. (C) Average gain magnitude on the BDI for two different treatments.

Limitations of the package include the fact that more substantial adaptations to the standard criteria cannot currently be implemented, though as the underlying code is publically available, researchers may wish to use this in combination with other tools for further development work. Second, while the package may significantly increase the speed and accuracy of calculations, it cannot and should not substitute considered methodological thinking. In particular, users should consider carefully the appropriateness of the methods selected within each function, including related assumptions and limitations. Lastly it should be emphasised that sudden gains and losses identified by applying a set of mathematical criteria are not necessarily

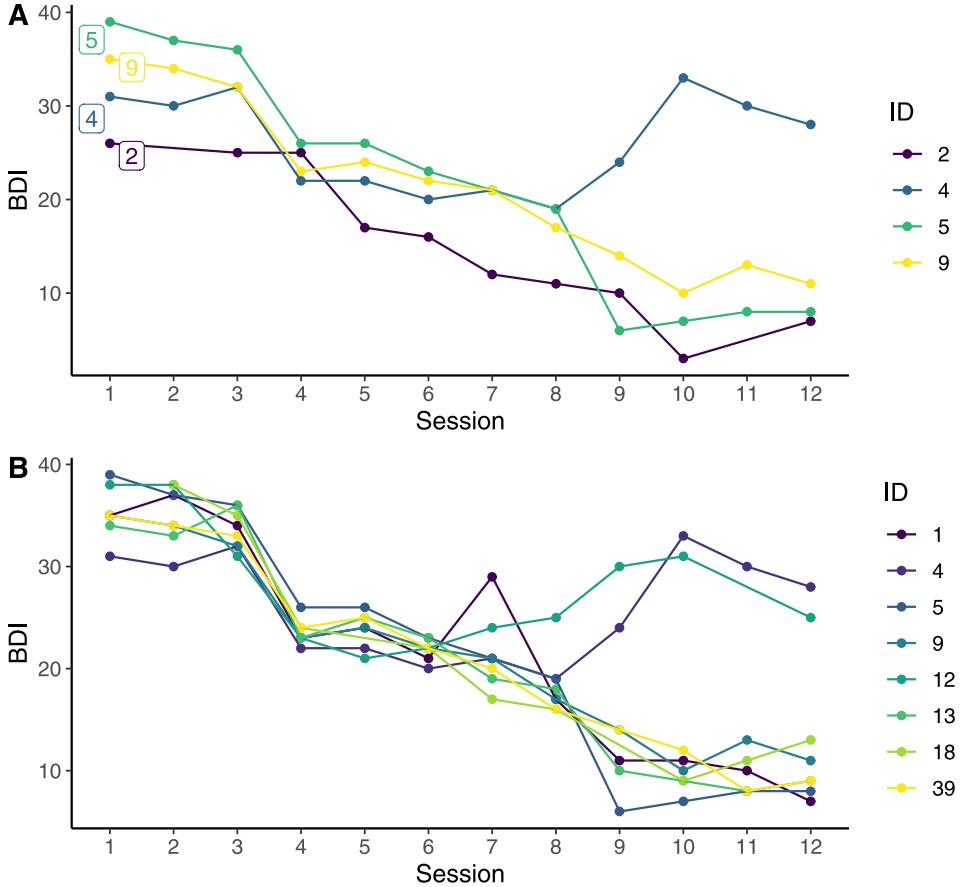

**Fig 2. Plots of trajectories for selected cases.** (A) Trajectories for a selection of individual cases. (B) Trajectories of BDI scores for all participants with a sudden gain between sessions 3 and 4.

related to the effects of the intervention being studied, and that further investigation would be required to establish the presence and strength of evidence for a causal relationship.

Overall, it is hoped that this package will permit faster and more transparent examination of sudden gains within a range of longitudinal datasets, and that it could provide a valuable tool to explore how the criteria might be refined or adapted to better identify gains that reflect meaningful change processes.

## Acknowledgments

Earlier versions of this manuscript were written using the R package *papaja* [29].

## Author Contributions

**Conceptualization:** Milan Wiedemann, Graham R. Thew, Richard Stott, Anke Ehlers.

**Funding acquisition:** Milan Wiedemann, Graham R. Thew, Anke Ehlers.

**Methodology:** Milan Wiedemann, Graham R. Thew, Richard Stott, Anke Ehlers.

**Project administration:** Milan Wiedemann.

**Software:** Milan Wiedemann.

**Supervision:** Anke Ehlers.

**Validation:** Milan Wiedemann, Graham R. Thew, Richard Stott.

**Visualization:** Milan Wiedemann, Graham R. Thew.

**Writing – original draft:** Milan Wiedemann, Graham R. Thew.

**Writing – review & editing:** Milan Wiedemann, Graham R. Thew, Richard Stott, Anke Ehlers.

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
