## [Decision Letter · Decision Letter 0]

19 Dec 2019

PONE-D-19-30704

suddengains: An R package to identify sudden gains in longitudinal data

PLOS ONE

Dear Mr Wiedemann,

Thank you for submitting your manuscript to PLOS ONE. After careful consideration, we feel that it has merit but does not fully meet PLOS ONE’s publication criteria as it currently stands. Therefore, we invite you to submit a revised version of the manuscript that addresses the points raised during the review process.

I have now received two reports by experts in the field and also read the manuscript myself. As you can see both reviewers evaluated your manuscript rather favorably and emphasized the important contribution of your work. At the same time, they also raised important shortcomings that need to be addressed before the manuscript can be accepted for publication. I will summarize the most important points and also include some points of my own:

You define sudden gains as “large and stable changes … between two consecutive measurement points” (page 2, line 2). From this introduction, it was unclear to me how you might infer conclusions about the stability of change if you have only two measurements. In my opinion, it would require at least three measurements to make conclusions about the extent the observed change is stable or not. On page 3, you modified this claim by evaluating six measurement points in criterion 3. I think the definition of sudden gains needs to be more precise and should be modified accordingly.Equation (1) uses the standard deviation of the difference score. This represents the formula for two independent groups. Although it is also sometimes used for dependent samples, Cohen (1988) suggested including the correlation between the two measures. In situations, when the standard deviation is expected to change (such as in intervention studies) Glass and colleagues (1981) even recommended using the standard deviation of the pre-intervention measurement (you even refer to this approach on page 4, line 63). Thus, there is some discussion on the appropriate way of estimating the variance of difference scores. Please include a more thorough discussion on your choice of the standard deviation and in what way it seems more appropriate than the other suggestions.

Cohen, J. (1988). Statistical Power Analysis for the Behavioral Sciences. New York, NY: Routledge Academic.

Glass, G. V., McGaw, B., and Smith, M. L. (1981). Meta-Analysis in Social Research. Beverly Hills, CA: Sage.

Reviewer 1 raises an important point regarding the assumptions in your approach (e.g., unidimensionality, constant reliabilities). I would recommend being more explicit about the inherent assumptions (and limitations) of you statistical model.Your way of handling missing data seems suspect to me (page 5). I doubt you can appropriately account for missingness simply by adapting you significance level (Reviewer 1 also raises this point). Please make sure to thoroughly describe the assumptions underlying this approach and also refer to Rubin’s typology of missing mechanisms (MCAR, MAR, MNAR).What are the criteria for excluding respondents because “it would not be possible to identify any sudden gains” (page 8, line 138)? Looking at Table 2, it seems to me that you considered one measurement point before and at least two measurement points after the pregain session as a necessary requirement. Could you elaborate why? Moreover, criterion 3 on page 3 seems to indicate that at least three measurements before and after the pregain session are needed?Reviewer 2 has two suggestions for improving the package and increasing its usefulness for applied researchers. You might want to consider these points. However, their implementation is not a requirement for the publication of your manuscript.

Your worked example is only partially presented. I would recommend also including the output generated by the functions and providing a step by step interpretation of the printout.

In addition to these points, both reviewers made further excellent suggestions that you should consider in your revision. I strongly encourage you to address these issues and submit a revised version of your manuscript.

We would appreciate receiving your revised manuscript by Feb 02 2020 11:59PM. To enhance the reproducibility of your results, we recommend that if applicable you deposit your laboratory protocols in protocols.io, where a protocol can be assigned its own identifier (DOI) such that it can be cited independently in the future. For instructions see: http://journals.plos.org/plosone/s/submission-guidelines#loc-laboratory-protocols

We look forward to receiving your revised manuscript.

Kind regards,

Timo Gnambs

Academic Editor

PLOS ONE

Journal Requirements:

**When submitting your revision, we need you to address these additional requirements:**

2. Please note that PLOS ONE has specific guidelines on software sharing (http://journals.plos.org/plosone/s/materials-and-software-sharing#loc-sharing-software) for manuscripts whose main purpose is the description of a new software or software package. In this case, new software must conform to the Open Source Definition (https://opensource.org/docs/osd) and be deposited in an open software archive.

Reviewers' comments:

Reviewer's Responses to Questions

**Comments to the Author**

1. Is the manuscript technically sound, and do the data support the conclusions?

Reviewer #1: Yes

Reviewer #2: Partly

2. Has the statistical analysis been performed appropriately and rigorously? 

Reviewer #1: Yes

Reviewer #2: N/A

3. Have the authors made all data underlying the findings in their manuscript fully available?

Reviewer #1: Yes

Reviewer #2: Yes

4. Is the manuscript presented in an intelligible fashion and written in standard English?

Reviewer #1: Yes

Reviewer #2: Yes

5. Review Comments to the Author

Reviewer #1: The manuscript “suddengains: An R package to identify sudden gains in longitudinal data” describes the development and use of a statistical package to identify sudden gains in psychotherapy. The rationale for the development of the package is provided and its features and use are discussed.

I believe that this manuscript is extremely important in the field of sudden gains research. As the authors noted, the application of sudden gains criteria has differed between studies, and the analyses and calculations are highly prone to errors. As a researcher in this field I think it is safe to assume that at least a moderate degree of variance and discrepancy between studies is due to applying criteria differently (e.g., changing vs. maintaining the critical value for the third criterion based on amount of missing data) and due to computational error. As a result, I found myself in complete agreement with the authors on the importance of having a standard package for computing sudden gains. I believe this can reduce some of the “error variance” that exists in the field.

As part of my review, I compared the suddengains package with a complex, multi-tab excel spreadsheet that I had been using for the past few years to identify sudden gains. The results were identical. While it is more likely that I validated my excel file with the suddengains package than validated the package with my excel file, I believe this suggests that the main functions work without error.

There are two potential features that I think could strengthen the package further. These are by no means necessary for publication as I believe the current features already represent a substantial contribution to the literature. However, if the authors have plans on updating the package in the future, I hope they consider these features. The first is the ability to chose the critical cutoff for the third criterion. As the authors noted, the third criterion is considered to be more a descriptive cutoff than a valid test of statistical significance. This has led some authors to change to the critical value to 2.5 for instance (Hardy et al., 2005). This option could be helpful in future versions. The second feature would be to allow for an altered 3rd criterion by using 1.5 standard deviations of symptom scores as a critical value. This has been done in many previous studies (e.g., Kelly et al. 2005; 2007). Allowing users to chose such an option could facilitate within-study comparisons of sudden gains identified using different criteria. I believe such a comparison could help the field decide on standard criteria based on empirical data.

Overall, I think the manuscript and package represent an important contribution to the field of sudden gains and believe the package will be used by many researchers in the field.

Reviewer #2: Review of the manuscript “suddengains: An R package to identify sudden gains in longitudinal data”

The manuscript focuses important information on individual change, which can be investigated in longitudinal data (i.e., sudden gains & losses). Central is the description and investigation of sudden gains. For the description of sudden gains, the authors consider specific identification criteria. Based on these criteria, they provide an R-package facilitating automated investigations of sudden gains. This should help to improve the efficiency, reporting, and reproducibility of sudden gains research.

I support the basic idea of the authors that the R-package “suddengains” is a useful toolbox for applied researches. In addition to a function for obtaining sudden gains, the presented R package has very helpful add-ons, like an illustrative example, a function for creating new output datasets with sudden gains, and graphical functions for illustrating sudden gains and individual trajectories. The authors build on available state-of-the-art packages, like dplyr and ggplot2, and their functions are well documented.

Despite my respect for developing such a useful R-package, I cannot recommend acceptance of the manuscript in the present form. Here is why: Every automation poses the risk of an unconsidered application of specific methods. In my view, this risk is not appropriately addressed in the manuscript, so far. The authors describe a very broad research field for investigating sudden gains with the presented package (p.2. line 8 – any longitudinal dataset with regular repeated measurement). I would encourage a much more detailed view on the context for the application of the R package, where the major limitations are:

1) The authors specify specific criteria for the identification of sudden gains, but they do not review the incorporated assumptions. For instance, calculating the standard error of measurement (S_E) in Eq. 6 or 7 with a specific reliability estimate (retest reliability or Chronbach’s alpha) is only possible, when investigating unidimensional scales, for which such reliability estimates hold. Furthermore, even when unidimensionality holds, the reliability of a measure is not necessarily constant over time or persons (see e.g., models of Latent-State-Trait Theory, where multiple time points are considered and Chronbach’s alpha can be calculated for unidimensional scales at each time point, or Item-Response-Theory models, which allow for heterogeneous measurement error variance depending on the ability of a person). Using one reliability measure for all time points and individuals is a rough approximation to control for measurement error and alternative methods are available. I think, more details on the assumptions of the promoted methods would be beneficial, in order to describe the limitations of the related R-package.

2) In the preparation of the data, the authors facilitate the exclusion of specific cases, when not enough data points are available to identify sudden gains. They clearly describe which cases have to be excluded. However, the assumptions for and the consequences of excluding persons are not mentioned. This is especially important when conducting subsequent analyses, which are encouraged by the presented R-package. When excluding persons with missing data that is not completely at random, then subsequent results can be biased.

3) A further (minor) comment addresses the statements on the meaning of sudden gains. The authors use a descriptive, databased view on sudden gains. This is appropriate for obtaining such events in the data. However, at the same time they address the meaning of sudden gains, for instance, on page 2 line 20-22 they refer to the meaning of sudden gains in placebo interventions. A clear meaning of sudden gains as effects of an intervention would require a more formal specification of sudden gains as causal effects of the intervention. Accordingly, further (design or analysis) conditions need to hold, for ruling out alternative explanations. I would clearly distinguish between obtaining sudden gains and their meaning.

To sum up, I encourage a revision of the manuscript - especially for the description of the methods and the discussion. Possible limitations of the methods and the related R package should be included. As such, the circumstances under which the application of the R-package is appropriate would be clearer. I would also appreciate a more detailed view on the meaning of sudden gains. In my view, the suggested changes can support reproducibility of sudden gains research, which is one of the author’s goals.

6. PLOS authors have the option to publish the peer review history of their article (what does this mean?). If published, this will include your full peer review and any attached files.

Reviewer #1: Yes: Idan M Aderka

Reviewer #2: No

---

## [Author Response · Author response to Decision Letter 0]

4 Feb 2020

Editor’s comments:

E#1. You define sudden gains as “large and stable changes … between two consecutive measurement points” (page 2, line 2). From this introduction, it was unclear to me how you might infer conclusions about the stability of change if you have only two measurements. In my opinion, it would require at least three measurements to make conclusions about the extent the observed change is stable or not. On page 3, you modified this claim by evaluating six measurement points in criterion 3. I think the definition of sudden gains needs to be more precise and should be modified accordingly.

RESPONSE: Thank you for raising this, having looked again we appreciate the wording was unclear - while ‘large and stable’ tends to be how gains are conceptualised in a broad sense, it is true that stability is evaluated not on 2 points, but on 6 as described later. We have amended the first sentence of the introduction to make this clearer (Page 2):

“A sudden gain is a large improvement in an outcome variable experienced by an individual participant between two consecutive measurement points that is stable within a longitudinal data series.”

E#2. Equation (1) uses the standard deviation of the difference score. This represents the formula for two independent groups. Although it is also sometimes used for dependent samples, Cohen (1988) suggested including the correlation between the two measures. In situations, when the standard deviation is expected to change (such as in intervention studies) Glass and colleagues (1981) even recommended using the standard deviation of the pre-intervention measurement (you even refer to this approach on page 4, line 63). Thus, there is some discussion on the appropriate way of estimating the variance of difference scores. Please include a more thorough discussion on your choice of the standard deviation and in what way it seems more appropriate than the other suggestions. 

Cohen, J. (1988). Statistical Power Analysis for the Behavioral Sciences. New York, NY: Routledge Academic.

Glass, G. V., McGaw, B., and Smith, M. L. (1981). Meta-Analysis in Social Research. Beverly Hills, CA: Sage.

RESPONSE: You rightly raise this issue, one that has been discussed in various sudden gains papers. Our use of the standard deviation is primarily driven by how Criterion 3 was originally defined by Tang and colleagues (1999, 2005, 2015) and has been subsequently applied in the literature. Our view is that for the R package to be useful it needs to be able to apply the criteria using the methods that are ‘standard’ for the field in order to allow comparability between studies. However, we also agree with the point raised by Reviewer 2 that we want to avoid the ‘unconsidered application’ of the package, so have edited the manuscript to make clearer the situations where there are methodological alternatives, and to make clearer references to studies that explore limitations and criticisms of the ‘standard’ approach. An empirical investigation comparing the standard approach with the Cohen and Glass methods would indeed be valuable for the field, and it is hoped that the open code of this package could facilitate such a study (note that we have recorded this as a suggested ‘issue’ on the package GitHub page, see https://github.com/milanwiedemann/suddengains/issues/23).

E#3. Reviewer 1 raises an important point regarding the assumptions in your approach (e.g., unidimensionality, constant reliabilities). I would recommend being more explicit about the inherent assumptions (and limitations) of you statistical model.

RESPONSE: Thank you - as described above we have examined the full manuscript in order to be more explicit in describing the relevant assumptions and limitations. We have improved the referencing of papers that discuss common limitations and relevant methodological issues. The assumptions of unidimensionality and constant reliabilities are now discussed on Page 5:

“Note that the use of the test-retest reliability or Cronbach’s alpha when calculating SE makes the assumption that the scale being examined is unidimensional, and that these reliability estimates remain constant over time, and between individuals. Researchers should consider exploring the factor structure and measurement invariance of the scale to examine if these assumptions hold.”

E#4. Your way of handling missing data seems suspect to me (page 5). I doubt you can appropriately account for missingness simply by adapting you significance level (Reviewer 1 also raises this point). Please make sure to thoroughly describe the assumptions underlying this approach and also refer to Rubin’s typology of missing mechanisms (MCAR, MAR, MNAR).

RESPONSE: We have made a number of changes to the section on missing data including making reference to reservations in the literature about the suitability of common replacement approaches for sudden gains research. It was not our intention to suggest that the adjustment of the critical value in Criterion 3 is sufficient to fully account for missingness - we have reworded this to be clear that this is one approach that has been suggested and used in the literature, but that further work to examine missing data within sudden gains research is warranted. 

As the classification of missing data types (e.g. MCAR, MAR, MNAR) will vary from study to study we have opted to make reference to Rubin’s work and encourage researchers to consider the reasons why data may be missing and what methods may need to be employed as a result. This applies principally to the identification of gains themselves, where replacing missing data might lead to identifying “false gains” and is not therefore appropriate. However, different methods for handling missing data may be appropriate for subsequent analyses, so we have now mentioned this at the end of the section.

“Missing data, for example where a participant does not provide data on one or more occasions, need to be considered carefully when identifying sudden gains for several reasons. Firstly, depending on the number and pattern of missing data points for an individual, it may not be possible to identify sudden gains, see Table 2. Specifically, in order to estimate the standard deviation values in criterion 3, at least two of the three measurements immediately prior to the gain must be present, as well as at least two of the three measurements immediately following the gain. Some researchers have suggested that methods used to replace missing values, such as last observation carried forward or multiple imputation, may not be appropriate when identifying sudden gains given the potential for additional gains to be detected based on data that were not provided by participants [20,21].

Secondly, where values are missing in the period around the potential sudden gain, two approaches have been described to evaluate the stability of the change. Following the updated version of the third criterion by Tang and colleagues [5,9] some studies have used a critical value of 2.776 across all session to session intervals to check the stability [17]. An alternative approach adjusts the critical values used in criterion 3 (see Eq 1) based on the data that were available in the period around the potential sudden gain [16]: Where no data are missing t(4;97.5%) > 2.776; where one datapoint is missing either before or after the gain t(3;97.5%) > 3.182; and where one datapoint is missing both before and after the gain t(2;97.5%) > 4.303. This method has been adopted in some subsequent studies [22,24].

It is important to understand the reasons for missing data and consider whether methods to handle missing data need to be employed both at the identification stage and in subsequent analyses [18,19]. Further research to examine the impact of missing data and different methods to handle missing data when identifying sudden gains would be beneficial.”

E#5. What are the criteria for excluding respondents because “it would not be possible to identify any sudden gains” (page 8, line 138)? Looking at Table 2, it seems to me that you considered one measurement point before and at least two measurement points after the pregain session as a necessary requirement. Could you elaborate why? Moreover, criterion 3 on page 3 seems to indicate that at least three measurements before and after the pregain session are needed?

RESPONSE: We have revised the explanation of the select_cases() function to better characterise its use. Some published studies have set a minimum number of datapoints that must be present in order to be included in the analysis, and others have analysed only cases with sufficient data for at least one session-to-session to be tested. The select_cases() function is optional and can be used to apply such conditions should researchers wish. The lowest number of datapoints for a single participant that could still show a sudden gain is four, provided they are arranged in one of the patterns shown in Table 2. We have revised Table 2 in order to improve clarity about these patterns, as well as making a more explicit statement about the minimum amount of data required (Page 9):

“For all three criteria to be applied there must be data present for at least two of the three data points prior to, and two of the three following, the interval to be examined.”

E#6. Reviewer 2 has two suggestions for improving the package and increasing its usefulness for applied researchers. You might want to consider these points. However, their implementation is not a requirement for the publication of your manuscript.

RESPONSE: Thank you, we are grateful for these helpful suggestions and have responded to the specific ideas below.

E#7. Your worked example is only partially presented. I would recommend also including the output generated by the functions and providing a step by step interpretation of the printout.

RESPONSE: We have added additional text and tables to provide explanations of the output at each stage (pages 10-12) so that the worked example is clearer to follow. 

In addition to these points, both reviewers made further excellent suggestions that you should consider in your revision. I strongly encourage you to address these issues and submit a revised version of your manuscript.

Reviewer 1: 

The manuscript “suddengains: An R package to identify sudden gains in longitudinal data” describes the development and use of a statistical package to identify sudden gains in psychotherapy. The rationale for the development of the package is provided and its features and use are discussed.

I believe that this manuscript is extremely important in the field of sudden gains research. As the authors noted, the application of sudden gains criteria has differed between studies, and the analyses and calculations are highly prone to errors. As a researcher in this field I think it is safe to assume that at least a moderate degree of variance and discrepancy between studies is due to applying criteria differently (e.g., changing vs. maintaining the critical value for the third criterion based on amount of missing data) and due to computational error. As a result, I found myself in complete agreement with the authors on the importance of having a standard package for computing sudden gains. I believe this can reduce some of the “error variance” that exists in the field.

As part of my review, I compared the suddengains package with a complex, multi-tab excel spreadsheet that I had been using for the past few years to identify sudden gains. The results were identical. While it is more likely that I validated my excel file with the suddengains package than validated the package with my excel file, I believe this suggests that the main functions work without error.

There are two potential features that I think could strengthen the package further. These are by no means necessary for publication as I believe the current features already represent a substantial contribution to the literature. However, if the authors have plans on updating the package in the future, I hope they consider these features. 

R1#1. The first is the ability to chose the critical cutoff for the third criterion. As the authors noted, the third criterion is considered to be more a descriptive cutoff than a valid test of statistical significance. This has led some authors to change to the critical value to 2.5 for instance (Hardy et al., 2005). This option could be helpful in future versions. 

RESPONSE: Thank you, we have implemented this function in a development version of the package and will add it into the main package after it is fully tested (see commit ce26a73, https://github.com/milanwiedemann/suddengains/commit/ce26a73ffb249c52c85e2c2d37546be3c59cd870, on the plos-one-revisions branch (see https://github.com/milanwiedemann/suddengains/tree/plos-one-revisions); Guidance on installing this developmental version can be found in the Readme file, see https://github.com/milanwiedemann/suddengains/tree/plos-one-revisions#installation). This function is also implemented in the interactive demonstration of the package, see https://milanwiedemann.shinyapps.io/shinygains/.

R1#2. The second feature would be to allow for an altered 3rd criterion by using 1.5 standard deviations of symptom scores as a critical value. This has been done in many previous studies (e.g., Kelly et al. 2005; 2007). Allowing users to chose such an option could facilitate within-study comparisons of sudden gains identified using different criteria. I believe such a comparison could help the field decide on standard criteria based on empirical data.

RESPONSE: Thank you for this suggestion. It raises the interesting topic of identifying sudden gains based on criteria that are different for each individual. We see this as a subsequent project that the package could facilitate - our view is that the development and implementation of these alternative methods would require some broader consideration and experimentation before they are ready to implement within the package, but that we would be keen to support this and to have such functions available in the package in future. We have added an issue describing this feature request on GitHub, see https://github.com/milanwiedemann/suddengains/issues/22. 

R1#3. Overall, I think the manuscript and package represent an important contribution to the field of sudden gains and believe the package will be used by many researchers in the field.

RESPONSE: Thank you for your kind comments and helpful suggestions.

Reviewer 2: 

The manuscript focuses important information on individual change, which can be investigated in longitudinal data (i.e., sudden gains & losses). Central is the description and investigation of sudden gains. For the description of sudden gains, the authors consider specific identification criteria. Based on these criteria, they provide an R-package facilitating automated investigations of sudden gains. This should help to improve the efficiency, reporting, and reproducibility of sudden gains research.

I support the basic idea of the authors that the R-package “suddengains” is a useful toolbox for applied researches. In addition to a function for obtaining sudden gains, the presented R package has very helpful add-ons, like an illustrative example, a function for creating new output datasets with sudden gains, and graphical functions for illustrating sudden gains and individual trajectories. The authors build on available state-of-the-art packages, like dplyr and ggplot2, and their functions are well documented.

R2#1. Despite my respect for developing such a useful R-package, I cannot recommend acceptance of the manuscript in the present form. Here is why: Every automation poses the risk of an unconsidered application of specific methods. In my view, this risk is not appropriately addressed in the manuscript, so far. The authors describe a very broad research field for investigating sudden gains with the presented package (p.2. line 8 – any longitudinal dataset with regular repeated measurement). I would encourage a much more detailed view on the context for the application of the R package, where the major limitations are:

RESPONSE: We share your concern regarding the ‘unconsidered application’ of the package and have therefore sought to address this risk within the manuscript. We have changed the mention of ‘any’ longitudinal dataset to suggest that other fields of research investigating within-participant changes may wish to consider this approach (Page 2, Line 7). We have included a number of additions throughout the paper to make readers more aware of the assumptions, limitations, and criticisms of the standard criteria and methods to identify sudden gains, and have improved the citation of relevant literature discussing these issues. These additions include specific statements encouraging the reader to consider their methodological choices carefully when using the package, e.g. Page 15:

“while the package may significantly increase the speed and accuracy of calculations, it cannot and should not substitute considered methodological thinking. In particular, users should consider carefully the appropriateness of the methods selected within each function, including related assumptions and limitations.”

On Page 8 we have also added a link to an interactive web-based ‘Shiny’ application we have been developing, that is designed to help readers of the paper and users of the package consider and understand the impact of different methodological choices (see https://milanwiedemann.shinyapps.io/shinygains/). We also note that the package itself provides a range of warning messages and help documentation to encourage careful and considered use of the functions. Responsibility does of course also lie with the end user, but we hope that the manuscript now better encourages considered use of the package functions.

R2#2. The authors specify specific criteria for the identification of sudden gains, but they do not review the incorporated assumptions. For instance, calculating the standard error of measurement (S_E) in Eq. 6 or 7 with a specific reliability estimate (retest reliability or Chronbach’s alpha) is only possible, when investigating unidimensional scales, for which such reliability estimates hold. Furthermore, even when unidimensionality holds, the reliability of a measure is not necessarily constant over time or persons (see e.g., models of Latent-State-Trait Theory, where multiple time points are considered and Chronbach’s alpha can be calculated for unidimensional scales at each time point, or Item-Response-Theory models, which allow for heterogeneous measurement error variance depending on the ability of a person). Using one reliability measure for all time points and individuals is a rough approximation to control for measurement error and alternative methods are available. I think, more details on the assumptions of the promoted methods would be beneficial, in order to describe the limitations of the related R-package.

RESPONSE: Thank you for highlighting this point. As noted above, we have now made a number of changes to the manuscript that aim to emphasise the assumptions and potential limitations of the methods that are used as the current ‘standard’ in the field. For the example you give above, we suspect it is true that most studies to date have used the ‘rough approximation’ approach to reliability that you describe, but of course agree it is important for researchers to understand and actively consider whether such assumptions are appropriate for their data. We have now included mention of this issue on Page 5:

“Note that the use of the test-retest reliability or Cronbach’s alpha when calculating SE makes the assumption that the scale being examined is unidimensional, and that these reliability estimates remain constant over time, and between individuals. Researchers should consider exploring the factor structure and measurement invariance of the scale to examine if these assumptions hold.”

In addition we have included clearer mention of one of the main limitations of the package in the discussion section (Page 15):

“Limitations of the package include the fact that more substantial adaptations to the standard criteria cannot currently be implemented, though as the underlying code is publically available, researchers may wish to use this in combination with other tools for further development work.”

R2#3. In the preparation of the data, the authors facilitate the exclusion of specific cases, when not enough data points are available to identify sudden gains. They clearly describe which cases have to be excluded. However, the assumptions for and the consequences of excluding persons are not mentioned. This is especially important when conducting subsequent analyses, which are encouraged by the presented R-package. When excluding persons with missing data that is not completely at random, then subsequent results can be biased.

RESPONSE: We have rephrased this section to avoid suggesting that certain participants need to be excluded, and instead clarify that the select_cases() function allows researchers to stipulate a minimum amount of data for each participant should they wish. In many cases, researchers may want to generate a dataset that consists only of participants with sudden gains (for example, to analyse characteristics of the gains themselves, or the participants who experience them), or one that also includes participants with sufficient data to demonstrate a gain but where one was not observed. In both scenarios, participants who have insufficient data to demonstrate a gain may therefore not be of interest. However, this will be determined by the research question and aims of the study, so we have emphasised that researchers should consider if this optional function is appropriate (Page 9; Please see also our response to comment E5).

“The optional select_cases() function can be used to identify samples of cases for analysis who fulfil such conditions, though researchers should consider whether these methods are appropriate for the aims of the study.”

R2#4. A further (minor) comment addresses the statements on the meaning of sudden gains. The authors use a descriptive, databased view on sudden gains. This is appropriate for obtaining such events in the data. However, at the same time they address the meaning of sudden gains, for instance, on page 2 line 20-22 they refer to the meaning of sudden gains in placebo interventions. A clear meaning of sudden gains as effects of an intervention would require a more formal specification of sudden gains as causal effects of the intervention. Accordingly, further (design or analysis) conditions need to hold, for ruling out alternative explanations. I would clearly distinguish between obtaining sudden gains and their meaning.

RESPONSE: Thank you for raising this important point. You rightly highlight the distinction between the data-driven, mathematical definition of a sudden gain, and their more conceptual meaning in relation to the intervention being studied. We have taken this opportunity to clarify wordings within the manuscript in order to better emphasise this difference, for example on Page 2:

“In addition, some studies have raised concerns about the validity of sudden gains identified through current methods, demonstrating that they can be found in placebo interventions and simulated datasets [3,4]. This suggests that not all gains reflect meaningful change or show a causal association with the intervention being studied. This highlights the need to examine the presence and strength of these associations and to consider if the current methods of identification can be refined.”

And in the discussion on Page 15:

“Lastly it should be emphasised that sudden gains and losses identified by applying a set of mathematical criteria are not necessarily related to the effects of the intervention being studied, and that further investigation would be required to establish the presence and strength of the evidence for a causal relationship.”

R2#5. To sum up, I encourage a revision of the manuscript - especially for the description of the methods and the discussion. Possible limitations of the methods and the related R package should be included. As such, the circumstances under which the application of the R-package is appropriate would be clearer. I would also appreciate a more detailed view on the meaning of sudden gains. In my view, the suggested changes can support reproducibility of sudden gains research, which is one of the author’s goals. 

RESPONSE: Thank you for your helpful comments. We hope we have been able to address them appropriately, and believe the manuscript has been much improved as a result.

---

## [Decision Letter · Decision Letter 1]

26 Feb 2020

suddengains: An R package to identify sudden gains in longitudinal data

PONE-D-19-30704R1

Dear Dr. Wiedemann,

We are pleased to inform you that your manuscript has been judged scientifically suitable for publication and will be formally accepted for publication once it complies with all outstanding technical requirements.

With kind regards,

Timo Gnambs

Academic Editor

PLOS ONE

Additional Editor Comments (optional):

Reviewers' comments:

Reviewer's Responses to Questions

**Comments to the Author**

1. If the authors have adequately addressed your comments raised in a previous round of review and you feel that this manuscript is now acceptable for publication, you may indicate that here to bypass the “Comments to the Author” section, enter your conflict of interest statement in the “Confidential to Editor” section, and submit your "Accept" recommendation.

Reviewer #2: All comments have been addressed

2. Is the manuscript technically sound, and do the data support the conclusions?

Reviewer #2: Yes

3. Has the statistical analysis been performed appropriately and rigorously? 

Reviewer #2: Yes

4. Have the authors made all data underlying the findings in their manuscript fully available?

Reviewer #2: Yes

5. Is the manuscript presented in an intelligible fashion and written in standard English?

Reviewer #2: Yes

6. Review Comments to the Author

Reviewer #2: The authors were very responsive to all suggestions and I found the revised paper very readable. In my view, the meaning of sudden gains and the limitations of standard research methods in this field are clearer. The R package “suddengains” will be very helpful for applied researchers. Furthermore, I support the authors’ statements that their open source code is beneficial for future methodological developments. Next to the good responses to my comments, I really liked the more detailed description of the example output in the manuscript as well as the add-on of a shiny app. Thus, I encourage acceptance of the manuscript.

7. PLOS authors have the option to publish the peer review history of their article (what does this mean?). If published, this will include your full peer review and any attached files.

Reviewer #2: Yes: Marie-Ann Sengewald

---

## [Editor Report · Acceptance letter]

28 Feb 2020

PONE-D-19-30704R1 

suddengains: An R package to identify sudden gains in longitudinal data 

Dear Dr. Wiedemann:

I am pleased to inform you that your manuscript has been deemed suitable for publication in PLOS ONE. Congratulations! Your manuscript is now with our production department. 

With kind regards,

on behalf of

Dr. Timo Gnambs 

Academic Editor

PLOS ONE